# Sister chromatid exchanges induced by perturbed replication can form independently of BRCA1, BRCA2 and RAD51

Anne Margriet Heijink[1,11,13], Colin Stok[1,12,13], David Porubsky[2,3], Eleni Maria Manolika[4], Jurrian K. de Kanter[5,6], Yannick P. Kok[1], Marieke Everts[1], H. Rudolf de Boer [1], Anastasia Audrey[1], Femke J. Bakker [1], Elles Wierenga[1], Marcel Tijsterman [7], Victor Guryev[2], Diana C. J. Spierings [2], Puck Knipscheer [6,8], Ruben van Boxtel [5,6], Arnab Ray Chaudhuri[4], Peter M. Lansdorp [2,9,10] ✉ & Marcel A. T. M. van Vugt [1] ✉

Sister chromatid exchanges (SCEs) are products of joint DNA molecule resolution, and are considered to form through homologous recombination (HR). Indeed, SCE induction upon irradiation requires the canonical HR factors BRCA1, BRCA2 and RAD51. In contrast, replication-blocking agents, including PARP inhibitors, induce SCEs independently of BRCA1, BRCA2 and RAD51. PARP inhibitor-induced SCEs are enriched at difficult-to-replicate genomic regions, including common fragile sites (CFSs). PARP inhibitor-induced replication lesions are transmitted into mitosis, suggesting that SCEs can originate from mitotic processing of under-replicated DNA. Proteomics analysis reveals mitotic recruitment of DNA polymerase theta (POLQ) to synthetic DNA ends. POLQ inactivation results in reduced SCE numbers and severe chromosome fragmentation upon PARP inhibition in HR-deficient cells. Accordingly, analysis of CFSs in cancer genomes reveals frequent allelic deletions, flanked by signatures of POLQ-mediated repair. Combined, we show PARP inhibition generates under-replicated DNA, which is processed into SCEs during mitosis, independently of canonical HR factors.

Double-stranded DNA breaks (DSBs) are toxic DNA lesions that can lead to cell death or genomic alterations if left unrepaired. Cells have evolved multiple DNA repair mechanisms to deal with DNA breaks[1,2]. In G1 phase of the cell cycle, DNA breaks are predominantly repaired through non-homologous end-joining (NHEJ), which involves ligation of DNA ends independently of sequence homology and frequently results in the introduction of small indels across the break site[3]. In contrast, when DNA has been replicated during S-phase, the sister chromatids can be used as templates for error-free repair through homologous recombination (HR)[4]. Cyclin-dependent kinase (CDK)-mediated activation of CtIP promotes the resection of the broken DNA ends by BRCA1/BARD1 and the MRE11/RAD50/NBS1 (MRN) complex,

generating single-stranded DNA (ssDNA) overhangs. The resected ends are poor substrates for NHEJ, marking a point of no return for initiation of HR. Subsequently, BRCA2 promotes loading of RAD51 recombinase onto ssDNA stretches[5,6]. RAD51 monomers are assembled into nucleoprotein filaments, which ultimately perform the homology search and invasion of the repair template[7,8]. Upon finding homology with the sister chromatid, DNA synthesis takes place via synthesis-dependent strand annealing (SDSA) or through the formation of a joint DNA molecule known as a Holliday junction (HJ). In order to allow faithful chromosome segregation, HJs need to be removed before the onset of mitosis[9]. HJs can be either 'dissolved' by the BLM/RMI1/RMI2/TopIIIa (BTR) complex or 'resolved' by the SLX1/SLX4/MUS81/EME1

---

complex or the GEN1 nuclease[9]. Upon completion of DNA synthesis by SDSA and after BTR-mediated dissolution of a HJ, the two ends of the DNA break are rejoined to the original sister chromatid, giving rise to a so-called 'non-crossover' event. Alternatively, when the DNA ends of opposing sister chromatids are rejoined, this results in a 'crossover' event or 'sister chromatid exchange (SCE)'[10–14]. Thus, whereas HJ dissolution exclusively gives rise to non-crossover events[15], HJ resolution can give rise to either non-crossover events or crossover end products.

Cells that lack functional BRCA1 or BRCA2, as for instance observed in hereditary breast or ovarian cancers, are defective in HR and display high levels of genomic instability[16–18]. Due to their DNA repair defect, HR-deficient cancer cells display enhanced sensitivity to DNA damaging agents, including DNA cross-linking agents such as cisplatin[19,20]. Particularly, HR-deficient cells are sensitive to inhibition of PARP1, an enzyme involved in DNA single-strand break repair[21,22]. The synthetic lethal interaction between BRCA deficiency and PARP1 inhibitors was initially explained by accumulation of single-strand DNA breaks due to PARP inhibition, which are converted into DSBs that are toxic in the absence of HR repair. However, PARP inhibitors also trap PARP molecules onto DNA[23]. PARP trapping induces stalling and collapse of replication forks[24,25], which are resolved, at least in part, by the HR machinery[26]. As a consequence, PARP inactivation leads to elevated levels of SCEs, which are considered products of HR[27–29].

The prevailing model in literature describes a requirement for HR components, including RAD51, in the formation of spontaneous and mutagen-induced SCEs[30,31]. Here we show that, in contrast to irradiation (IR)-induced lesions, DNA lesions induced by replication-blocking agents, including PARP inhibitors, give rise to under-replicated DNA regions in mitosis, which are processed into SCEs independently of canonical HR.

## Results

### Olaparib induces sister-chromatid exchanges in BRCA2-proficient and deficient cancer cells

To study the effects of PARP inhibition on the induction of sister-chromatid exchanges (SCEs), we employed Strand-seq[32,33], which allows single cell sequencing of the DNA template strand and genome-wide mapping of SCEs[34]. Murine tumor-derived $Tp53^{-/-}Brca2^{-/-}$ KB2P3.4 cells and $Brca2$-reconstituted $Tp53^{-/-}Brca2^{IBAC}$ KB2P3.4R3 cells[35] were treated with the PARP inhibitor olaparib, and libraries were prepared for Strand-seq analysis (Fig. 1A). A significant increase in the number of SCEs was observed in $Brca2$-proficient KB2P3.4R3 cells in response to PARP inhibitor treatment (Fig. 1A, B), in line with previous findings[29,36]. Surprisingly however, Strand-seq analysis revealed that olaparib treatment induced a comparable number of SCEs in $Brca2$-deficient KB2P3.4 cells (Fig. 1A, B). Of note, a significant increase in copy number variations (CNVs) was observed in $Brca2$-deficient KB2P3.4 cells (Fig. 1C and Supplementary Fig. 1A), a phenotype consistent with HR-deficiency[16]. Using differential "harlequin" chromatid staining of metaphase spreads, olaparib-induced SCEs were again observed in the absence of $Brca2$, validating our previous observation (Fig. 1D, E). Moreover, background levels of SCEs were similar in $Brca2$-proficient and $Brca2$-deficient cells, indicating that spontaneous SCEs, like olaparib-induced SCEs, arise independently of $Brca2$ (Fig. 1B, E). Thus, both spontaneous and PARP inhibitor-induced SCEs arise in $Brca2$-deficient cells, showing that $Brca2$ is not essential for SCE formation.

### Olaparib-induced SCEs arise independently of canonical HR factors in a dose-dependent manner

To investigate whether BRCA2-independent SCEs can be observed in other cell lines, we introduced shRNAs targeting BRCA2 in untransformed human $TP53^{-/-}$ RPE-1 cells (Fig. 2A). We observed a loss of irradiation-induced RAD51 foci in BRCA2-depleted $TP53^{-/-}$ RPE1 cells (Supplementary Fig. 2A)[37], validating functional loss of HR in these cells. Moreover, we analyzed replication forks by electron microscopy,

and observed a significant increase in the amounts of ssDNA gaps in BRCA2-depleted cells upon PARP inhibition (Fig. 2B), in line with recent reports[38–41]. Notably, EM analysis also revealed complex 'branched' replication structures, predominantly in BRCA2-depleted cells upon olaparib treatment (Supplementary Fig. 2B, C).

In line with defective HR, BRCA2-depleted $TP53^{-/-}$ RPE1 cells showed increased sensitivity to PARP inhibitor treatment (Supplementary Fig. 3A). When RPE-1 cells were treated with sublethal concentrations of olaparib (Supplementary Fig. 3A), we again observed a significant and dose-dependent increase in the number of SCEs (Fig. 2C, Supplementary Fig. 3B), which occurred independently of BRCA2 (Fig. 2C). Interestingly, induction of SCEs by γ-irradiation (IR) was completely dependent on BRCA2 (Fig. 2D), in line with previous reports[42]. To test whether olaparib-induced SCEs also arise independently of other canonical HR components, we depleted BRCA1, an upstream HR regulator, and RAD51, the main recombinase responsible for strand invasion (Fig. 2A). In line with our results in BRCA2-depleted cells, olaparib-induced SCEs were also observed in cells expressing shRNAs against BRCA1 or RAD51 (Fig. 2C). In stark contrast, IR-induced SCEs were fully dependent on BRCA1 and RAD51 (Fig. 2D), indicating that the processing of IR-induced and olaparib-induced lesions show different dependencies on canonical HR factors. Furthermore, proteins involved in the resolution of joint molecules downstream in the HR pathway, including MUS81, ERCC1 and SLX1-SLX4 were also not required for the formation of olaparib-induced SCEs in RPE-1 cells, whereas these proteins were required for IR-induced SCEs (Supplementary Fig. 3C–E).

Since RAD51 has a central role in recombination, SCE formation was assessed in a $RAD51^{-/-}$ DT40 chicken B-cell lymphoma cell line as well, which depends on the expression of a doxycycline-repressible hRAD51 transgene for viability[43]. Treatment with doxycycline resulted in robust repression of hRAD51 expression (Supplementary Fig. 3F), allowing us to study RAD51-dependent events in these cells. Metaphase spreads revealed high numbers of gaps and breaks in these cells, characteristic of RAD51 deficiency (Supplementary Fig. 3G). Notably, and in accordance with our findings in RPE-1 cells, olaparib treatment in these cells induced formation of SCEs independently of RAD51 (Fig. 2E), whereas IR-induced SCEs were fully dependent on RAD51 (Fig. 2F). In parallel, we used the small molecule inhibitor B02 to inhibit the DNA strand exchange activity of RAD51[44], and again observed SCE induction upon olaparib treatment independently of RAD51 (Fig. 2G). By contrast, RAD51 inhibition completely prevented SCE induction in IR-treated cells (Fig. 2H). Overall, these data show that olaparib-induced SCEs are independent of canonical HR components in multiple HR-deficient cell line models.

### Induction of HR-independent SCEs by replication-perturbing agents

In addition to inhibiting the enzymatic activity of the PARP1/2 enzymes, most PARP inhibitors, including olaparib, are capable of trapping PARP to the DNA[23,45]. To distinguish between the effects of PARP inhibition and PARP trapping, siRNAs against PARP1 were introduced in RPE-1 cells (Fig. 3A). siRNA-mediated depletion of PARP1 induced significantly less SCEs when compared to PARP inhibition (Fig. 3B), suggesting that PARP trapping is the predominant source for the generation of HR-independent SCEs. To test if PARP-trapping agents other than olaparib also induce SCEs, we treated RPE-1 cells with a panel of PARP inhibitors that trap PARP onto DNA with variable capacity[23]: olaparib, veliparib and talazoparib (Figs. 3B and S3A). When used at IC25 dose, all three PARP inhibitors induced SCEs, which were again generated independently of BRCA2 (Fig. 3B). Since PARP-trapping inhibitors have been reported to disturb normal replication fork progression[25,46,47], we investigated whether SCEs also arise upon treatment with other replication-perturbing agents, to exclude effects of the roles of PARP1/2 in DNA repair. A panel of commonly used

chemotherapeutics that target DNA replication was tested, including the DNA crosslinking agents mitomycin C (MMC) and cisplatin, and the topoisomerase inhibitors camptothecin (CPT) and etoposide. Although these agents induced SCEs to various extents, very similar amounts of SCEs were observed in control and BRCA2-depleted cells (Fig. 3B). Only for etoposide, a small but significant decrease in SCEs

was observed in BRCA2-depleted cells (Fig. 3B). Overall, our data suggests that the capacity to block DNA replication forks is a key determinant in the formation of HR-independent SCEs.

To investigate whether PARP inhibitor-induced SCEs are associated with specific genomic features, we performed Strand-seq in olaparib-treated KBM-7 cells, expressing control or BRCA2-targeting

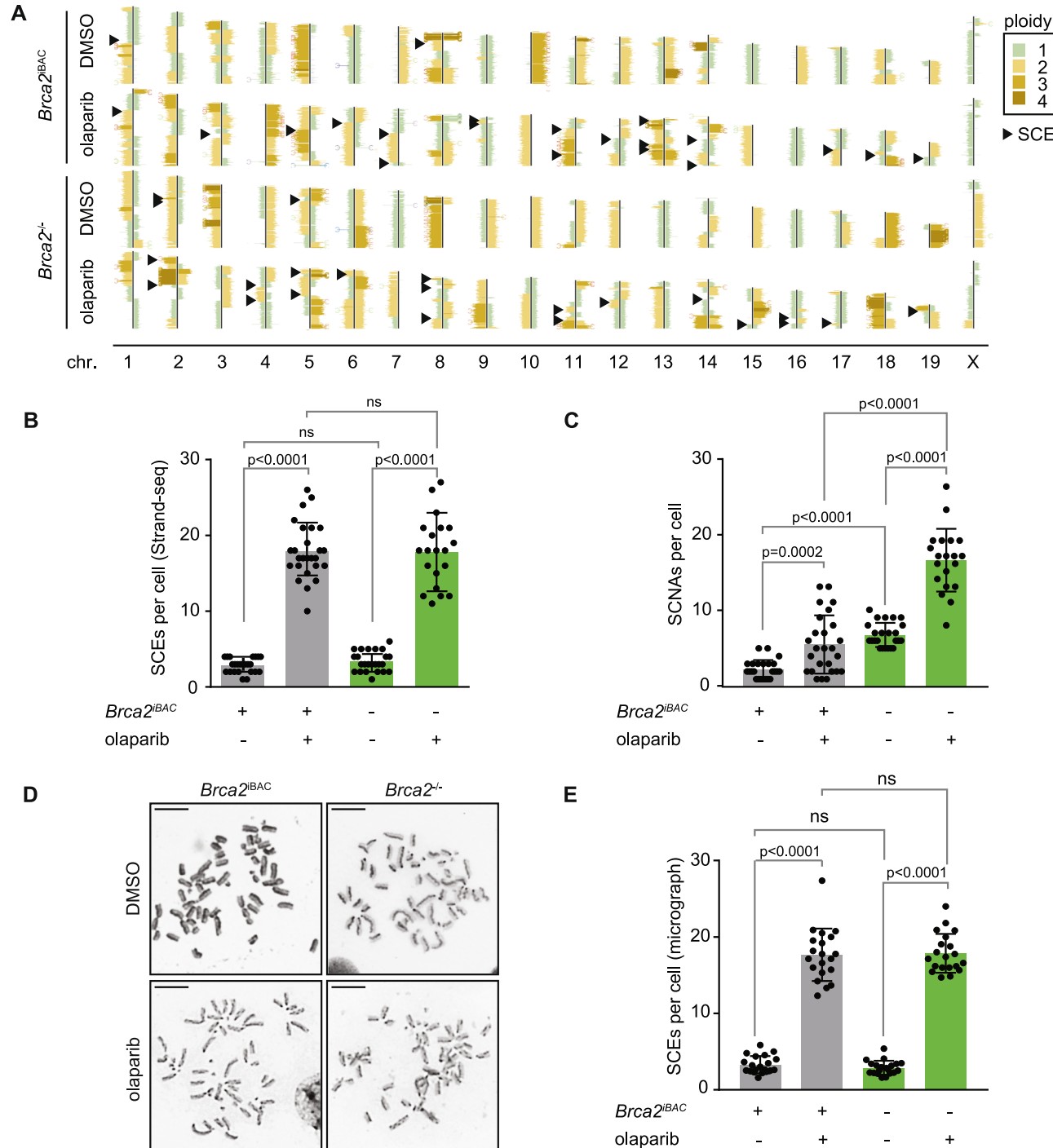

**Fig. 1 | Olaparib-treatment induces sister-chromatid exchanges in *Brca2* wt and *Brca2*-mutant cancer cells. A** Representative Strand-seq libraries of *Brca2*⁻/⁻ cells or *Brca2*^iBAC cells, treated with DMSO (top) or olaparib (bottom). Black arrowheads indicate SCEs. **B, C** *Brca2*⁻/⁻ cells or *Brca2*^iBAC cells were treated with DMSO or olaparib. Quantification of SCEs (panel B) and CNVs (Panel C) per cell was done in *n* = 23 (*Brca2*^iBAC, DMSO), *n* = 26 (*Brca2*^iBAC, olaparib), *n* = 24 (*Brca2*⁻/⁻, DMSO), and *n* = 19 (*Brca2*⁻/⁻, olaparib) libraries. Means and standard deviations

are indicated. **D, E** *Brca2*⁻/⁻ cells or *Brca2*^iBAC cells were treated with DMSO or olaparib, and SCEs were quantified by microscopy analysis of *n* = 20 metaphase spreads per condition (**D**). Scale bars indicate 10 μm. Averages and standard deviation are presented (**E**). For panels **B, C, E** statistics were performed using unpaired two-tailed *t*-tests (ns: non-significant). Gray bars indicate HR-proficient conditions, green bars indicate HR-defective conditions. Source data are provided with this paper.

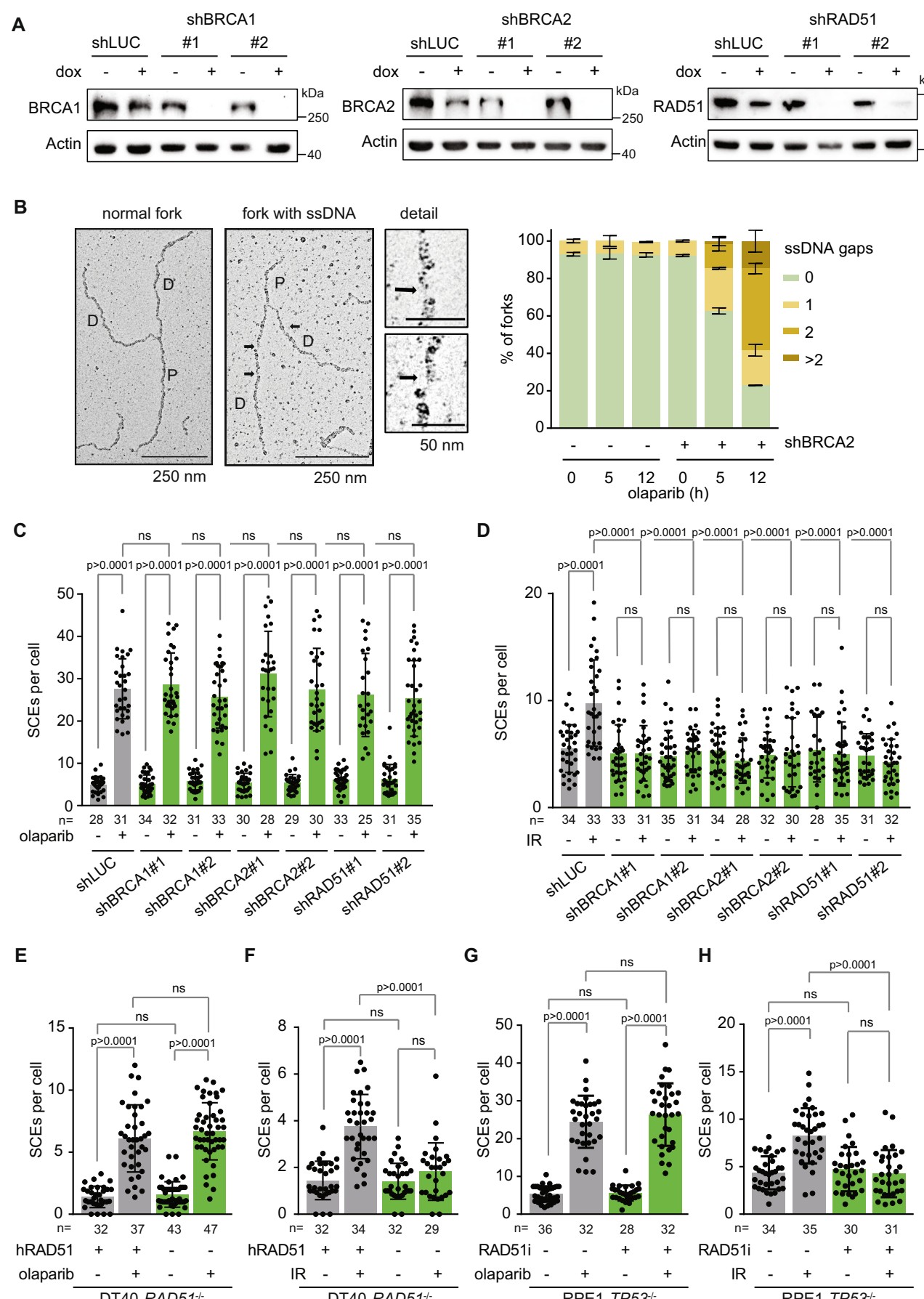

**Fig. 2 | HR-independent induction of SCEs upon PARP inhibitor treatment.**
**A** RPE-1-*TP53*[-/-] cells with indicated dox-inducible shRNAs were treated with doxycycline for 48 h and immunoblotted for indicated proteins. Data is representative for one biologically independent experiment (**B**) RPE1 *TP53*[-/-] shBRCA2 cells were pre-treated for 48 h with doxycycline (dox) and treated with olaparib for indicated time periods. Representative electron microscopy images are indicated of normal replication forks, and replication forks with ssDNA gaps. Quantification of ssDNA gaps is presented in the right panel. Averages and standard deviations of 60 replication forks per condition are shown. **C, D** RPE-1-*TP53*[-/-] cells with indicated dox-inducible shRNAs were treated with doxycycline for 48 h, and subsequently treated with olaparib for 48 h (**C**) or 2 Gy irradiation (**D**). SCEs were quantified by microscopy analysis of at least 25 metaphase spreads per condition from one biologically independent experiment, with exact n values indicated in the figure.

Means and standard deviations are plotted. **E, F** DT40 *RAD51*[-/-] cells harboring a dox-repressed hRad51 transgene were treated with doxycycline for indicated time periods, and treated with olaparib (**E**) or irradiation (**F**). SCEs in macrochromosomes were quantified by microscopy analysis of at least 29 metaphase spreads per condition from one biologically independent experiment. Exact n values are indicated in the figure. Means and standard deviations are plotted. **G, H** RPE1-*TP53*[-/-] cells were incubated with olaparib (**G**) or IR (**H**) in the absence or presence of the RAD51 inhibitor BO2. SCEs were quantified by microscopy analysis of at least 28 metaphase spreads per condition from one biologically independent experiment. Exact *n* values are indicated in the figure. Means and standard deviation are plotted. Statistics in panels **B**, **C**, **E**, **F**, **G**, **H** were performed using unpaired two-tailed *t*-tests (ns: non-significant). Gray bars indicate HR-proficient conditions, green bars indicate HR-defective conditions. Source data are provided with this paper.

shRNAs. The near haploid karyotype of KBM-7 cells allows for robust mapping of SCEs. In line with our previous observations, PARP inhibition induced SCEs in KBM-7 cells independently of BRCA2 (Supplementary Fig. 4A). KBM-7 cells displayed deletions, amplifications and copy number variations in BRCA2-depleted cells, which were further increased upon PARP inhibition, underscoring a functional HR defect in these cells (Supplementary Fig. 4B–D). SCE locations were mapped using HapSCElocatoR, as described previously (Supplementary Fig. 4E)[48]. Genomic locations of SCEs were mapped against the locations of previously described human CFSs[49]. A significant enrichment of olaparib-induced SCEs was observed within CFS regions in BRCA2-deficient cells (Fig. 3C, D and Supplementary Data 1), in line with CFSs being regarded as difficult-to-replicate loci. For example, in BRCA2-deficient cells treated with olaparib, a substantial number of SCEs were detected within the FRA1B and FRA4F CFSs (Fig. 3C). Subsequently, SCEs were mapped against centromeres and telomeres to test enrichment at other difficult-to-replicate regions (Supplementary Fig. 4F, G). Whereas no SCEs were observed at telomeric regions, olaparib also significantly induced centromeric SCEs in both BRCA2-deficient and control cells (Supplementary Fig. 4F, G). Moreover, BRCA2-depleted cells showed a significant depletion of SCEs within gene bodies (Supplementary Fig. 4H). Finally, no significant enrichments were observed at putative G4 structures (Supplementary Fig. 4I). The observation that SCEs in BRCA2-deficient cells are enriched at CFSs and centromeric regions is in accordance with our hypothesis that these SCEs are associated with DNA replication fork stalling.

To further investigate the relation between DNA replication and SCE formation, RPE-1 cells were treated with olaparib for different time periods, treating cells either during or after S-phase (Fig. 3E, F). EdU incorporation was used to assess the time required for RPE-1 cells to progress from S-phase to mitosis (Fig. 3E). After 8 h of EdU incorporation, only 4.9% of all mitoses (phospho-histone H3-positive cells) had incorporated EdU, suggesting that the majority of mitotic cells did not progress through S-phase at the time of collection (Fig. 3E). Accordingly, 8 h olaparib treatment induced only a minor number of SCEs (Fig. 3F). In contrast, 12 or 24 h EdU treatment resulted in a considerable population of EdU-positive mitotic cells (Fig. 3E), which coincided with significantly larger numbers of SCEs being induced (Fig. 3F). Overall, these data suggest that olaparib needs to be present during S-phase in order to induce SCEs in the following mitosis. Combined, these findings illustrate that PARP inhibition in HR-deficient cells leads to extensive replication perturbation, and that HR-independent SCEs are observed in a range of conditions, with perturbed replication as a shared mechanism-of-action.

### Processing of olaparib-induced DNA lesions during mitosis

We previously reported that olaparib-induced replication lesions are transmitted into mitosis[25,50]. To further test whether olaparib treatment induces mitotic DNA lesions in BRCA2-deficient RPE-1 *TP53*[-/-] cells, we measured the DNA damage markers γH2AX and FANCD2 in mitotic cells

(Fig. 4A). Both endogenous and olaparib-induced γH2AX and FANCD2 foci were enriched in BRCA2-depleted cells (Fig. 4A). Since mitotic FANCD2 foci reflect the presence of unresolved under-replicated DNA, we next assessed if olaparib treatment leads to increased mitotic DNA synthesis (MiDAS). We observed an increase in mitotic EdU foci in response to olaparib treatment in BRCA2-depleted cells (Fig. 4B), suggesting that DNA replication in these cells is incomplete at the moment of mitotic entry. Although MiDAS was most prominently induced in BRCA2-depleted cells, we also observed a mild increase in MiDAS foci in control cells treated with olaparib (Fig. 4B). To further investigate the link between mitosis and SCE formation, RPE-1 cells were treated with an inhibitor of the ATR checkpoint kinase to force cells into mitosis with under-replicated DNA (Fig. 4C). Cells treated with ATR inhibitor showed elevated numbers of SCEs, independently of BRCA2 (Fig. 4D). Similarly, inhibition of the cell cycle checkpoint kinase Wee1 also resulted in premature mitotic entry (Fig. 4E), along with elevated numbers of BRCA2-independent SCEs (Fig. 4F). Combined, these observations suggest that mitotic processing of under-replicated DNA may be the source for HR-independent SCEs.

Recently, it was hypothesized that SCEs could also originate from mitotic processing of stalled replication forks[51]. This would involve cleavage of both leading or both lagging strands, introducing DSBs surrounding the under-replicated genomic region. Since canonical NHEJ is inactivated during mitosis[52,53], we searched for proteins that act on DNA breaks during mitosis. To this end, we used biotin-tagged synthetic DNA structures that resemble DNA double-strand breaks, and pulled out associated proteins from interphase or mitotic *Xenopus laevis* egg extracts (Fig. 5A and Supplementary Data 2). Mass spectrometry analysis revealed a range of proteins that were enriched on synthetic DNA ends in mitotic extracts, including cip2a and dpolq, the *Xenopus* orthologs of CIP2A and TOPBP1 (Fig. 5B), which together with MDC1 are involved in mitotic tethering of mitotic DNA breaks[54,55]. Intriguingly, we also identified the single-strand annealing (SSA) factor RAD52 and the alternative end-joining (alt-EJ) factor POLQ (Fig. 5B). Inactivation of RAD52 in RPE-1 cells using CRISPR/Cas9 (Supplementary Fig. 5A), either alone or in combination with BRCA2 depletion, did not decrease SCEs amounts in olaparib-treated cells (Fig. 5C), suggesting that SSA through RAD52 is not responsible for PARP inhibitor-induced SCEs. Rather, RAD52 inactivation resulted in an increase in PARP inhibitor-induced SCEs, suggesting that RAD52 acts as an inhibitor of mitotic SCE formation. In contrast, upon CRISPR/Cas9-mediated inactivation of POLQ (Supplementary Fig. 5B), a decrease in SCEs was observed in BRCA2-deficient RPE-1 cells (Fig. 5C). Moreover, we observed severe fragmentation of mitotic chromosomes, consistent with an inability to process mitotic breaks resulting from under-replicated DNA (Fig. 5D). Chromosome fragmentation was also observed in BRCA2-deficient cells treated with the recently described POLQ inhibitor novobiocin[56] (Supplementary Fig. 5C, D). Moreover, POLQ inhibition resulted in a minor extension of mitotic duration of BRCA2-deficient cells, suggesting a function for POLQ in the repair of DNA lesions during mitosis (Fig. 5E).

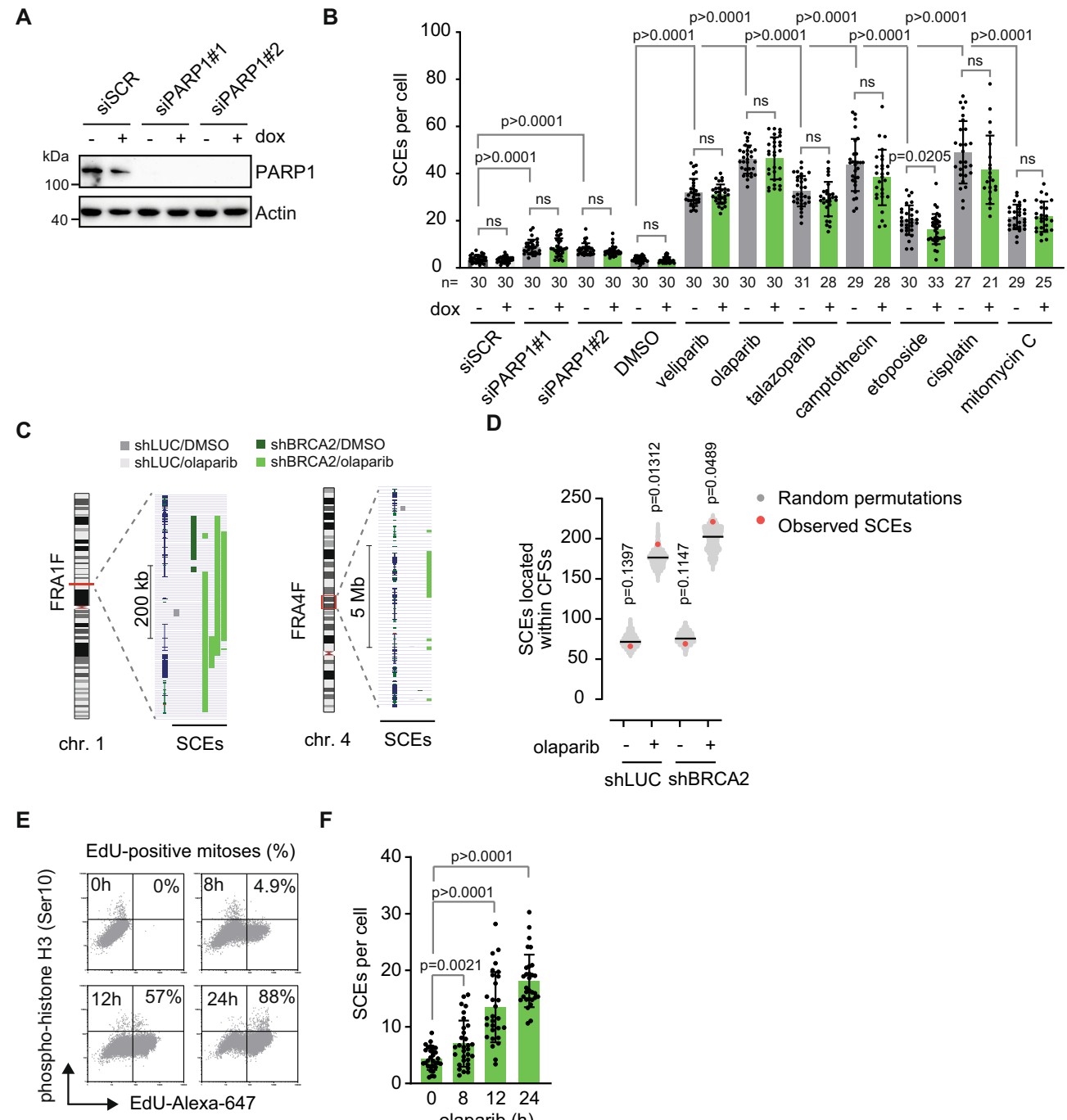

**Fig. 3 | HR-independent SCEs are associated with defective replication. A** RPE-1-*TP53⁻ᐟ⁻* shBRCA2 cells were pre-treated with doxycycline (dox) and transfected with siPARP or control siRNAs. Lysates were immunoblotted for PARP1 and Actin. Data are representative for two independent experiments. **B** RPE-1-*TP53⁻ᐟ⁻* shBRCA2 cells were pre-treated with doxycycline (dox) and subsequently treated with indicated agents for 48 h or transfected with PARP1 siRNAs 48 h before harvest. SCEs were quantified by microscopy analysis of at least 21 metaphase spreads per condition from one biologically independent experiment. Exact *n* values are indicated in the graph. Averages and standard deviations are indicated. **C**, **D** KBM-7 cells harboring doxycycline-inducible control or BRCA2 shRNAs were pre-treated with doxycycline and subsequently treated with olaparib where indicated. SCEs were mapped using StrandSeq of *n* = 64 (shLUC/DMSO), *n* = 31 (shLUC/OLA), *n* = 50 (shBRCA2/DMSO), and *n* = 52 (shBRCA2/OLA) libraries per condition from one biologically independent experiment. Observed SCEs were mapped to CFSs. Statistical analysis was

performed using one-sided permutation with 10,000 iterations (**D**). *P* values indicate deviation of the observed number of SCEs compared to the mean of all permutations. SCE mapping to the common fragile sites FRA1F and FRA4F are presented as illustrative examples (**C**). **E**, **F** Doxycycline pre-treated RPE-1 *TP53⁻ᐟ⁻* shBRCA2 cells were treated with ethynyl deoxyuridine (EdU) for the indicated time periods, and subsequently analyzed for mitotic cells by flow cytometry of phospho-histone H3 (Ser10). Percentages of mitotic cells that were EdU-positive are indicated (**E**). Doxycycline pre-treated RPE-1 *TP53⁻ᐟ⁻* shBRCA2 cells were treated with olaparib for the indicated time points, and SCEs were quantified by microscopy analysis of 30/31/29/30 metaphase spreads per condition from one biologically independent experiment (**F**). Means and standard deviations are plotted. Statistics in panels **B** and **F** were performed using unpaired two-tailed *t*-tests (ns: non-significant), and gray bars indicate HR-proficient conditions, green bars indicate HR-defective conditions. Source data are provided with this paper.

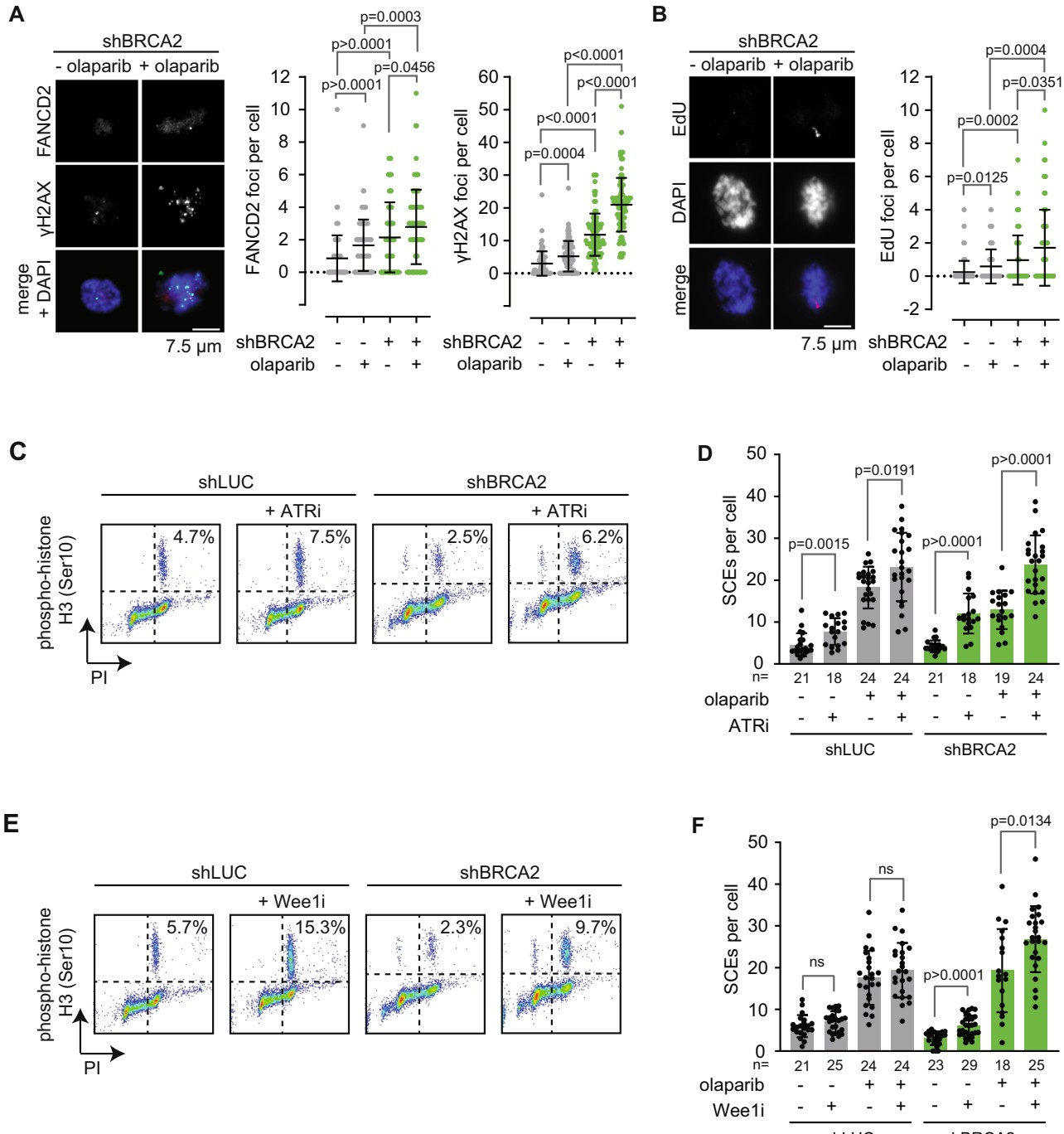

**Fig. 4 | SCEs originate from mitotic processing of under-replicated DNA. A** RPE1 *TP53*<sup>-/-</sup> shBRCA2 were pre-treated with doxycycline (dox), synchronized using RO-3306 for 4 h, and subsequently treated with olaparib where indicated. γH2AX and FANCD2 foci in mitotic cells were quantified by immunofluorescence microscopy. Means and standard deviation of pooled data from three independent experiments are shown, with *n* = 30 mitoses per experiment. **B** RPE1 *TP53*<sup>-/-</sup> shBRCA2 were treated as for panel **A**. 24 h after olaparib treatment, cells were incubated for 25 min with EdU. Mitotic EdU foci were quantified in *n* = 30 mitoses per experiment. Means and standard deviation of pooled data from three independent experiments are shown. **C**, **D** RPE1 *TP53*<sup>-/-</sup> shBRCA2 cells were treated with doxycycline (dox) for 48 h, with olaparib for 24 h, with or without the ATR inhibitor VE-821 (ATRi) for 3 h. Cells were treated with colcemid for 3 h before harvesting, fixed and stained for the

mitotic marker phospho-Histone-H3 (**C**). In parallel, SCEs were quantified by microscopy analysis of at least 18 metaphase spreads per condition. Exact *n* values are indicated in the figure (**D**). Exact *n* values are provided in the figure. **E**, **F** RPE1 *TP53*<sup>-/-</sup> shBRCA2 cells were treated with doxycycline (dox) for 48 h, with olaparib for 24 h, with or without the Wee1 inhibitor AZD-1775 (Wee1i) for 3 h. Cells were treated with colcemid for 3 h before harvesting, fixed and stained for the mitotic marker phospho-Histone-H3 (**E**). In parallel, SCEs were quantified by microscopy analysis of at least 18 metaphase spreads per condition. Exact n values are indicated in the figure (**F**). Statistics in panels **A**, **B**, **D**, and **F** were performed using two-sided Mann-Whitney tests (ns: non-significant). Gray bars indicate HR-proficient conditions, green bars indicate HR-defective conditions. Source data are provided with this paper.

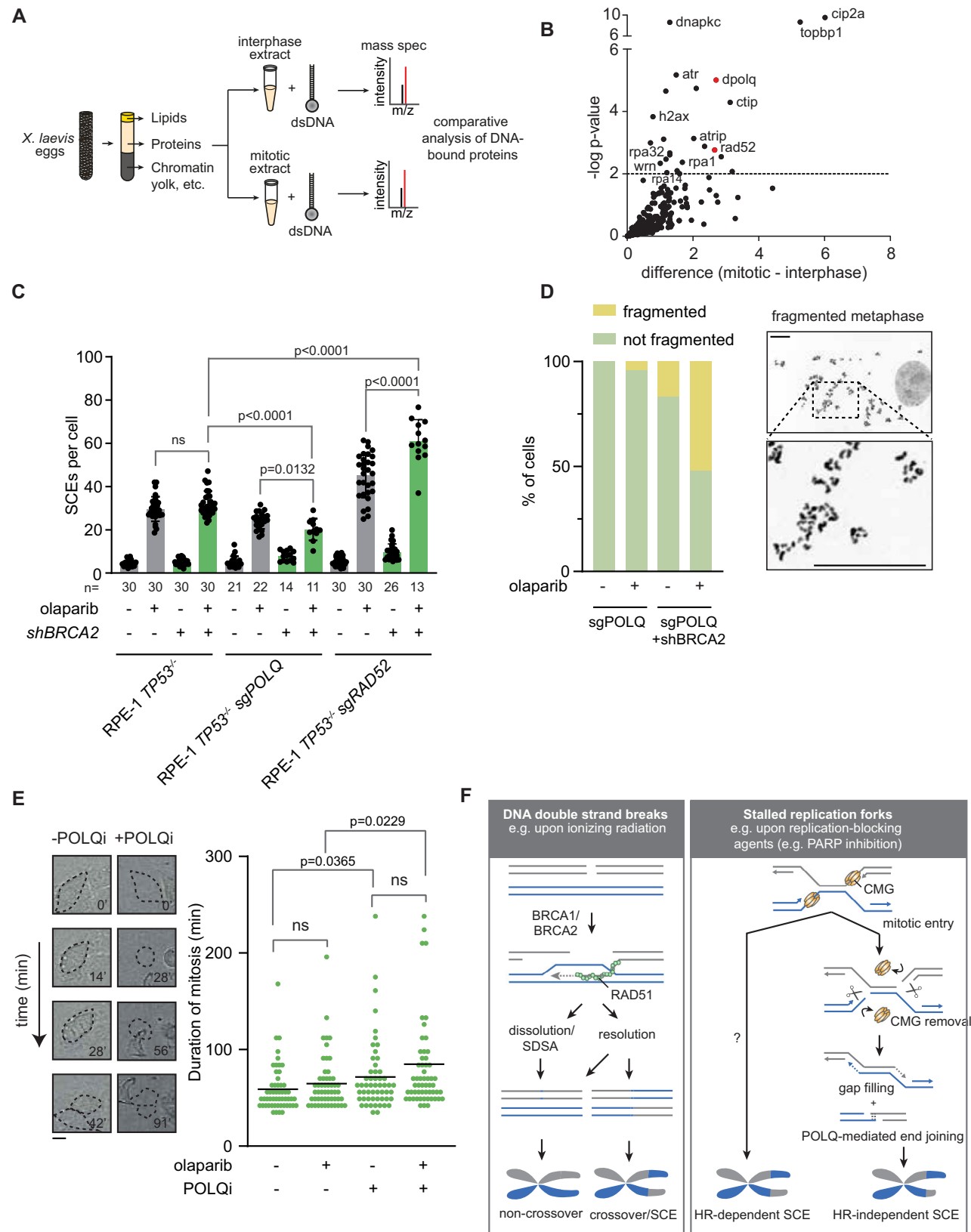

Our data fit a model in which transmission of under-replicated DNA into mitosis results in SCE formation that can arise in a HR-independent fashion (Fig. 5F). We hypothesize that upon removal of the replisome at the onset of mitosis, DNA breaks are induced, flanking the under-replicated DNA, consistent with previously reported data[51]. Subsequently, the ssDNA gaps flanking the under-replicated region are filled, and ligation of the two broken sister chromatids is promoted by POLQ (Fig. 5F). As a consequence of mitotic processing of under-replicated DNA according to this model, an HR-independent SCE is formed, which is predicted to be accompanied by allelic deletions with a size reflecting the extent of under-replication (Fig. 5F). To test this model, we analyzed whole-genome sequencing (WGS) data of a cohort

**Fig. 5 | SCEs originate from mitotic processing of under-replicated DNA.**
**A**, **B** Interphase or mitotic *Xenopus* egg extracts were prepared and incubated with biotin-conjugated blunt-ended DNA oligos (**A**). Proteins associated with DNA oligos were identified by mass spectrometry (**B**). *P* values were calculated using two-sided unpaired Student's T tests, with equal variance and a false discovery rate of 0.01. **C** RPE1 *TP53*[-/-] sgPOLQ shBRCA2 or RPE1 *TP53*[-/-] sgRAD52 shBRCA2 cells were pre-treated with doxycycline (dox) and subsequently treated with olaparib where indicated. SCEs were quantified by microscopy analysis of 11–30 mitoses per condition (exact n values are indicated in the figure) from one biologically independent experiment. Means and standard deviations are indicated. **D** RPE1 *TP53*[-/-] sgPOLQ shBRCA2 cells were treated as for panel **C**, and fragmented DNA was analyzed for 50 mitoses per condition from one biologically

independent experiment. Scale bars indicate 5 µm. **E** DIC images of RPE1 *TP53*[-/-] shBRCA2 cells were obtained every 7 minutes for 10 h. Cells were treated with olaparib and POLQi where indicated. Representative images and quantification of mitotic duration are shown for n = 54/54/54/57 cells from one biologically independent experiment. Scale bar indicates 10 µm. **F** Model of HR-independent SCE formation. Left panel indicates HR-dependent SCE formation after DSB repair. Right panel indicates HR-independent SCE formation by mitotic processing of under-replicated DNA. Statistics in panel **C** were performed using unpaired two-tailed *t*-tests. Statistical analysis in panel E was done using a two-sided Mann–Whitney test (ns: non-significant). Gray bars indicate HR-proficient conditions, green bars indicate HR-defective conditions. Source data are provided with this paper.

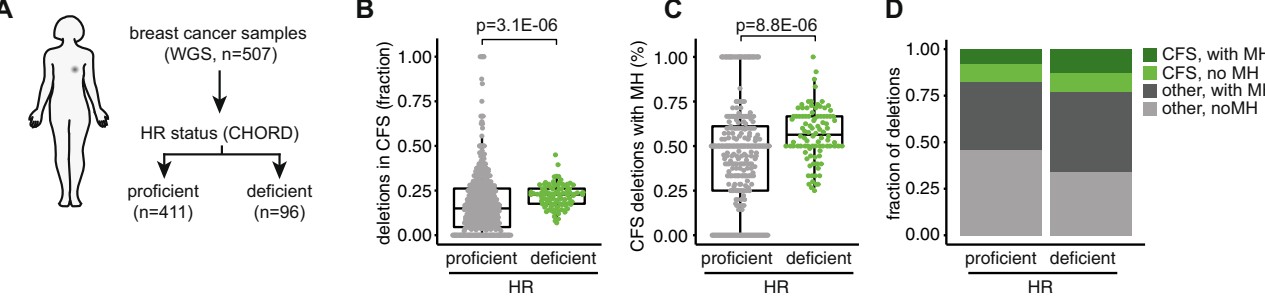

**Fig. 6 | Analysis of genomic deletions at common fragile sites. A** Whole-genome sequence (WGS) data of *n* = 507 breast cancers of the International Cancer Genome Consortium (ICGC) were analyzed, of which *n* = 96 of which were HRD. HRD status was assessed using the CHORD algorithm. **B**, **C** Allelic deletions from 507 breast cancers mapping to common fragile sites were plotted in panel **B**. Allelic deletions mapping to common fragile sites and having microhomology (MH)

(larger or equal to 2 bp) at the breakpoints were plotted in panel **C**. Box plots depict the mean (center line), 25th and 75th percentiles (box boundaries), and the largest values no more than 1.5* the interquartile range (whiskers). Statistical analysis was done using an FDR-corrected two-sided Wilcox test. **D** Analysis of deletions within CFS with or without MH as a percentage of total deletions in 507 breast cancers were plotted.

of 507 breast cancers. Since SCEs occur stochastically, bulk sequencing data cannot directly assess scars at spontaneous SCEs. Instead, we assessed CFSs, as these loci are enriched for SCEs (Fig. 3C, D). We separately analyzed HR-proficient and HR-deficient breast cancers, as assessed by the CHORD algorithm[57] (Fig. 6A), or by *BRCA1/2* mutation status (Supplementary Fig. 6A). We frequently observed allelic deletions that are positioned in CFSs, which was more frequently observed in HR-deficient cancers when compared to HR-proficient cancers (19% vs 15%, *p* = 7.55E[-11]; Fig. 6B), and more frequently observed in *BRCA1/2* mutant cancers (Supplementary Fig. 6B). Deletions frequently spanned mega-base regions (Supplementary Fig. 6C), in line with the previously reported length of under-replicated areas[58]. As HR-deficient tumors have previously been characterized by deletions flanked by ≥2 bp microhomology[57,59], we next tested whether allelic deletions at CFSs displayed microhomology at the break sites (Fig. 6C, D and Supplementary Fig. 6D, E), which reflects usage of polymerase theta-mediated end joining[60]. Indeed, a large fraction of deletions at CFSs showed 2 bp microhomology, with more CFS deletions in HR-deficient tumors harboring breakpoints with microhomology when compared to HR-proficient tumors (56% vs 47%, *p* = 8.8E[-6]). Similar results were obtained when comparing *BRCA1/2* mutant with *BRCA1/2* wildtype cancers (*p* = 5.4E[-4]; Fig. 6C and Supplementary Fig. 6D).

Combined, our data suggest that HR-independent SCEs can originate from mitotic processing of under-replicated DNA, and suggests the involvement of the alternative end-joining polymerase POLQ, leading to allelic loss of under-replicated loci.

## Discussion

We here show that agents that perturb DNA replication, including PARP inhibitors, induce sister chromatid exchanges in the absence of canonical HR factors BRCA1, BRCA2, and RAD51. Conversely, these HR components are required for induction of SCEs upon irradiation. Our

findings challenge the current dogma that SCEs solely arise as a result of homologous recombination[30,31,61–63]. Interestingly, models in which formation of SCEs upon replication fork stalling occur independently of HR have been proposed in the past[64], although these have lost support in favor of the HR-dependent models over the years. The observation that SCEs form independently of RAD51 is of particular interest. SCEs are considered to involve the formation of joint molecules, which typically requires strand invasion by RAD51. Previously observed replication stress-associated SCEs in *BRCA2* mutant cells were explained by BRCA2-independent RAD51 recruitment[65]. Yet, spontaneous and replication-induced SCEs have been reported frequently in cells lacking canonical HR factors[31,66–68], underscoring the notion that SCEs can arise independently of HR, and suggesting that PARP-inhibitor induced SCEs and spontaneous SCEs may share common mechanisms. Surprisingly, loss of the RAD51 paralogs RAD54, RAD51C, RAD51D, XRCC2, and XRCC3 has previously been reported to reduce MMC-induced and spontaneous SCEs, although these effects were limited and were attributed to defective RAD51 functioning[31,68].

Our data point towards incompletely replicated DNA as the source of DNA lesions that cause HR-independent SCEs. We and others find that PARP inhibition perturbs DNA replication in HR-defective cells[25,37,69], and that this leads to elevated levels of mitotic FANCD2 foci, a previously established marker of under-replicated DNA. Moreover, we observed moderate MiDAS activation in BRCA2-proficient cells, and further elevated levels of MiDAS in BRCA2-depleted cells upon PARP inhibition. Increased MiDAS was observed upon PARP inhibition in both BRCA2-depleted cells and control-depleted cells, although it occurred more frequently in BRCA2-depleted cells. MiDAS is commonly observed in situations of perturbed DNA replication, and may reflect an attempt of cells to finalize stalled DNA replication at the G2/M transition[58,70]. Importantly, incomplete DNA replication results in sister chromatids that are connected as 'joint DNA molecules' (Fig. 5F)

and would explain how SCEs could arise independently of strand invasion via BRCA1, BRCA2, and RAD51.

It is currently unclear if the ssDNA gaps that we observed at replication forks (Fig. 2B), play a role in the formation of HR-independent SCEs. The enrichment of both olaparib-induced SCEs and ssDNA gaps in BRCA2-deficient cells, warrants further research into potential shared mechanisms between SCE and ssDNA gap formation at stalled forks. If SCE formation at CFSs is directly linked to stalled replication forks, this would suggest that SCEs at CFSs may arise through different mechanisms than SCEs elsewhere in the genome. ssDNA gaps in BRCA1/2-deficient cell have previously been attributed to PRIMPOL activity[71]. Analysis of PRIMPOL-deficient cells may shed light on the role of ssDNA gap formation in in HR-independent SCEs.

Of note, differential BrdU staining protocols for SCE detection require scoring of mitotic chromosome spreads. A limitation of this method is that cells with extensive DNA damage will likely activate a G2 cell cycle checkpoint and fail to enter mitosis. By selectively analyzing mitotic cells, we may be looking at an underestimation of the amount of DNA lesions in these cells. Yet, the cells that do enter mitosis clearly demonstrate that SCEs can arise independently of canonical HR components.

In good agreement with our data, coordinated cleavage of stalled replication forks in mitosis was recently hypothesized to yield SCEs, at the cost of local deletions[51,72]. This process requires that the stalled replication forks flanking the under-replicated DNA are cleaved either at both leading strands, or at both lagging strands (Fig. 5E). Coordinated cleavage of replication forks could be initiated by TRAIP/p97-dependent unloading of the CMG helicase[51], leaving stretches of vulnerable ssDNA at the leading strands. Although the responsible nuclease in processing under-replicated DNA upon PARP inhibition remains elusive, MUS81 has been shown to be active during mitosis[11,73–75], to localize to under-replicated DNA in mitosis[76], and to act on stalled replication forks in mitosis[77,78]. However, our data showed that depletion of MUS81 alone was not sufficient to reduce SCEs, possibly due to redundancy with other endonucleases that are active during mitosis[11,79].

Cleavage and end-joining of cleaved under-replicated DNA regions during mitosis would yield SCEs as well as allelic deletions of under-replicated genomic regions. We report a potential role for POLQ in this process. Recently, POLQ was shown to act on DNA lesions during mitosis[80], although a role of POLQ in processing of mitotic DNA lesions into SCEs was not previously demonstrated to our knowledge. Of note, we do not find a full loss of HR-independent SCEs in POLQ-deficient cells. We therefore cannot rule out that other parallel pathways can mediate HR-independent SCEs. In contrast to our findings, Polq loss in mouse embryonic fibroblasts (MEFs) resulted in increased numbers of MMC-induced SCEs, although these SCEs may be a product of canonical HR in these cells[81].

The allelic deletions that are predicted in our model to arise during processing of under-replicated DNA during mitosis have been reported previously at common-fragile sites[82]. Moreover, the observed submicroscopic deletions that span CFS regions were shown to be flanked by microhomology regions, suggesting the involvement of POLQ-mediated end joining, consistent with the mutational signatures observed in BRCA1/2-deficient tumors[83,84]. Moreover, a role for POLQ in the mitotic processing of stalled replication forks in BRCA1/2-deficient cells fits well with the previously described synthetic lethality between POLQ and BRCA2[85–87], and between POLQ and other regulators of DSB repair[81]. Also, a model in which POLQ-mediated processing of under-replicated DNA functions in parallel to a pathway involving HR explains the observed synthetic lethal effects of POLQ inhibition in BRCA1/2-deficient cells, particularly in the context of PARP-inhibitor treatment[88].

Interestingly, our observation that mitotic SCEs arise independently of RAD52 implicates that we are not looking at a direct end-product of break-induced replication (BIR)[89,90] or MiDAS[70,91], which are both dependent on RAD52. Interestingly, MiDAS involves nuclease activity, and this pathway could therefore potentially compete with POLQ-dependent mitotic SCE formation. Indeed, RAD52 inactivation resulted in higher SCE numbers, suggesting that RAD52 inhibits SCE formation during mitosis. These findings are in good agreement with a recently described interaction between RAD52 and POLQ, in which RAD52 blocks POLQ activity during mitosis[80]. How MiDAS-mediated joint molecules can be formed during mitosis, when multiple nucleases are active to resolve such structures remains unclear. Interestingly, MiDAS was recently suggested to reflect completion of DNA replication at the end of G2 phase, rather than during mitosis[92], suggesting that MiDAS and mitotic SCE formation could be consecutive processes.

Although the synthetic lethal interaction between HR loss and PARP inhibition has been validated in various models and has been successfully exploited in the clinic, the fate of PARP inhibitor-induced DNA lesions in HR-deficient cells remains unclear[93]. Although we find that PARP inhibitor-induced DNA lesions are processed into SCEs in HR-deficient cells, we predict that this goes along with accumulation of large deletions and translocations due to mitotic POLQ activity, which may underlie loss of viability observed in these cells. In this context, a requirement for mitotic replisome unloading by TRAIP/p97, fits well with observed sensitization of HR-deficient cells for PARP inhibition upon inactivation of TRAIP[94], and the observation that progression through mitosis promotes PARP inhibitor-mediated cell death[25]. Our observation that PARP inhibitor-induced DNA lesions in HR-deficient cells are transmitted into mitosis also aligns well with the recent identification of the DNA tethering factor CIP2A being essential in HR-deficient cells[55], and was identified in our proteomics analysis of mitotic factors that bind DNA ends. Further research is warranted to investigate whether tethering of DNA ends is required to ligate DNA ends upon cleavage of stalled replication forks during mitosis.

## Methods

### Cell lines

hTERT-immortalized human retina epithelial RPE-1 cells and HEK293T cells were obtained from ATCC, and were maintained in Dulbecco's Minimum Essential Media (DMEM, Thermofisher), supplemented with 10% fetal calf serum (FCS, Lonza), 50 units/mL penicillin and 50 μg/mL streptomycin (P/S, Gibco) at 37 °C, 20% $O_2$ and 5% $CO_2$. KBM-7 cells were a kind gift from Thijn Brummelkamp (The Netherlands Cancer Institute, Amsterdam, The Netherlands) and were maintained in Iscove's Modified Dulbecco's Media (IMDM, Thermofisher) supplemented with 10% FCS and P/S. DT-40 cells were a kind gift from Shunichi Takeda (Kyoto University, Japan) and were grown in Roswell Park Memorial Institute (RPMI)-1640 media supplemented with 10% FCS, 1% chicken serum (Sigma) and P/S (Gibco) at 39.5 °C, 20% $O_2$ and 5% $CO_2$. The KB2P3.4 and KB2P3.4R3 cell lines were a kind gift from Jos Jonkers (The Netherlands Cancer Institute, Amsterdam, The Netherlands). The KB2P3.4 cell line was established from a mammary tumor from K14cre;Brca2$^{F11/F11}$;p53$^{F2-10/F2-10}$ mice and described previously[95]. The KB2P3.4R3 cell line was created by the stable introduction of an integrative bacterial artificial chromosome (iBAC), containing the full-length mouse Brca2 gene, into the KB2P3.4 cell line[95]. KB2P3.4 and KB2P3.4R3 cells were cultured in DMEM/F-12 medium, supplemented with 10% FCS, 50 units/mL penicillin, 50 μg/mL streptomycin, 5 μg/mL insulin (Sigma), 5 ng/mL epidermal growth factor (Life Technologies) and 5 ng/mL cholera toxin (Gentaur), at 37 °C and hypoxic conditions (1% $O_2$, 5% $CO_2$).

### Knockdown and knockout cell line models

To generate RPE-1 and KBM-7 cell lines expressing doxycycline-inducible short-hairpin RNAs (shRNAs), DNA oligos were cloned into Tet-pLKO-puro (Addgene plasmid #21915) vector. Tet-pLKO-puro was

a kind gift from Dmitri Wiederschain. shRNAs directed against luciferase ('shLUC', 5′-AAGAGCTGTTTCTGAGGAGCC-3′), BRCA2 (#1: 5′-GAAGAATGCAGGTTTAATA-3′ and #2: 5′-AACAACAATTACGAACCAAACTT-3′), BRCA1 (#1: 5′-CCCACCTAATTGTACTGAATT-3′ and #2: 5′-GAGTATGCAAACAGCTATAAT-3′), RAD51 (#1: 5′-CGGTCAGAGATCATACAGATT-3′ and #2: 5′-GCTGAAGCTATGTTCGCCATT-3′), MUS81 (#1 5′-GAGTTGGTACTGGATCACATT-3′ and #2 5′-CCTAATGGTCACCACTTCTTA-3′), SLX4 (#1 5′-ATTTCTGCTTCATTCACGTTT-3′ and #2 5′-CACCTGCAGACTCAAATGCCG-3′), and ERCC1 (#1 5′-CCAAGCCCTTATTCCGATCTA-3′ and #2 5′-CAAGAGAAGATCTGGCCTTAT-3′) were cloned into the Tet-pLKO-puro vector. Lentiviral particles were produced as described previously[96]. In brief, HEK293T packaging cells were transfected with 4 μg of indicated pLKO plasmid in combination with the packaging plasmids lenti-VSV-G and lenti-ΔVPR using a standard calcium phosphate protocol[97]. Virus-containing supernatant was harvested at 48 and 72 h after transfection and filtered through a 0.45 μM syringe filter. Supernatants were used to infect target cells in medium with a final concentration of 4 μg/mL polybrene (Sigma Aldrich). RPE-1 cells harboring a *TP53* mutation were generated by introducing a single-guide RNA (sgRNA) targeting exon 4 of the *TP53* gene as described previously[98]. To generate RAD52 and POLQ knockout cells, sgRNAs targeting exon 3 of RAD52 (AGAATACATAAGTAGCCGCA) and exon 1 of POLQ (GCCGGGCGGCGGGCTCAGCA) were cloned into the PX458 vector, which was a gift from Feng Zhang (Addgene plasmid # 48138). POLQ sgRNAs were a kind gift from Marcel Tijsterman (Leiden University Medical Centre, Leiden, the Netherlands). Plasmids were introduced in RPE-1 cells using Fugene HD transfection reagent and cells were selected based on GFP-expression or using 7 μg/mL puromycin (Sigma Aldrich) for 5 days.

### siRNA transfection

Cells were transfected with 40 nM siRNAs (Ambion Stealth RNAi, Thermofisher) targeting PARP1 (sequence 1: #HSS100243 and sequence 2: #HSS100244) or a scrambled (SCR) control sequence (sequence #12935300) with oligofectamine (Invitrogen), according to the manufacturer's recommendations.

### Sister chromatid exchange assays

SCE assays were performed as described previously[99]. RPE-1 cells were pre-treated with 0.1 μg/mL doxycycline (Sigma) for 48 h, followed by 48 h treatment with 10 μM BrdU. For BRCA2-deficient RPE-1 cells, BrdU treatment was increased to 64 h. Inhibitors were added for 48 h, simultaneously with BrdU treatment at the following concentrations: 0.5 μM olaparib (Axon Medchem), 16 μM veliparib (Axon Medchem), 7 nM talazoparib (Axon Medchem), 50 nM mitomycin C (Sigma), 5 μM cisplatin (Accord), 5 nM campthothecin (Sigma), 250 nM etoposide (Sigma), 20 μM BO2 (Axon Medchem), and 50 μM novobiocin (POLQi; Sigma-Aldrich). VE-821 (ATRi; Axon Medchem) or AZD-1775 (Wee1i; Axon Medchem) were added simultaneously with colcemid for 3 h at a concentration of 1.0 μM. Alternatively, cells were treated with 2 Gy γ-irradiation 8–10 h prior to fixation using an IBL 637 Cesium137 γ-ray source. Cells were collected in 10 μg/mL colcemid (Roche) for 3–6 h, fixed in 3:1 methanol:acetic acid solution and inflated in a hypotonic 0.075 M KCl solution. Metaphase spreads were made by dripping the cell suspensions onto microscope glasses from a height of ~30 cm. Slides were stained with 10 μg/mL bis-Benzimide H 33258 (Sigma) for 30 min, exposed to 245 nM UV light for 30 min, incubated in 2x SSC buffer (Sigma) at 60 °C for 1 h, and stained in 5% Giemsa (Sigma) for 15 min. DT40 cells were treated with BrdU for 48 h, doxycycline for 24 h and 0.5 μM olaparib for 24 h. Alternatively, DT40 cells were treated with BrdU and doxycycline as stated above, irradiated with 4 Gy and fixed at 8 h later. For DT40 cells, only macrochromosomes were included for analysis. KB2P3.4R3 cells were treated with BrdU for 32 h and KB2P3.4 for 40 h. Both KB2P3 cells lines were treated with 1 μM olaparib for 48 h.

### Immunofluorescence microscopy

RPE-1 cells were seeded on glass coverslips in 6-well plates and treated with doxycycline (1 μg/ml) and olaparib (0.5 μM). Cells were then treated for 4 h with the CDK inhibitor RO-3306 (5 μM). Upon washout of RO-3306, cells were incubated with EdU (20 μM) for 25 min. Cells were fixed using 2% formaldehyde in 0.1% Triton X-100 PBS for 10 minutes and subsequently permeabilized for 10 min in PBS with 0.5% Triton X-100. Staining was performed using primary antibodies against FANCD2 (Novusbio, Centennial, CO, USA; NB100-182, 1:200) and γH2AX Millipore, 05-636, 1:200). Cells were then incubated with corresponding Alexa-488 or Alexa-647-conjugated secondary antibodies and counterstained with DAPI (Sigma). For analysis of DNA damage response components, prophase and prometaphase cells were identified based on condensed chromatin conformation, and included for analysis. Images were acquired on a Leica DM6000B microscope using a ×63 immersion objective (PL S-APO, numerical aperture: 1.30) with Las-af software (Leica, Wetzlar, Germany).

For RAD51 analysis, RPE1 cells were left untreated or were irradiated using a Cesium137 source (CIS international/IBL 637 irradiator, dose rate: 0.01083 Gy per second). After 3 h, cells were washed in phosphate-buffered saline (PBS) and then fixed in 2% paraformaldehyde with 0.1% Triton X-100 in PBS for 30 min at room temperature. Cells were permeabilized in 0.5% Triton X-100 in PBS for 10 min. Subsequently, cells were extensively washed and incubated with PBS containing 0.05% Tween-20 and 4% bovine serum albumin (fraction V) (PBS-Tween-BSA) for 1 h to block nonspecific binding. Cells were incubated overnight at 4 °C with primary antibodies targeting RAD51 (GeneTex, GTX70230, 1:400). Cells were extensively washed and incubated for 1 h with Alexa-conjugated secondary antibodies (1:400) and counterstained with 4′,6-diamidino-2-phenylindole (DAPI). Slides were mounted with ProLong Antifade Mountant (Thermofisher). Images were acquired on a Leica DM-6000RXA fluorescence microscope, equipped with Leica Application Suite software.

### Cell viability assays

RPE-1 cells were plated in 96-wells plates at a concentration of 800 cells per well. After 24 hours, cells were treated with indicated concentrations of olaparib, veliparip, or talazoparib (all from Axon Medchem) for 3 days. Methyl-thiazol tetrazolium (MTT, Sigma) was added to cells at a concentration of 5 mg/mL for 4 hours, after which culture medium was removed and formazan crystals were dissolved in DMSO. Absorbance values were determined using a Bio-Rad benchmark III Biorad microtiter spectrophotometer at a wavelength of 520 nm.

### Western blot analysis

Cells were lysed in Mammalian Protein Extraction Reagent (MPER, Thermo Scientific), supplemented with protease inhibitor and phosphatase inhibitor (Thermo Scientific). Protein concentrations were measured using a Bradford assay. Proteins were separated by SDS-PAGE gel electrophoresis, transferred to Polyvinylidene fluoride (PVDF, immobilon) membranes and blocked in 5% skimmed milk (Sigma) in TRIS-buffered saline (TBS) containing 0.05% Tween-20 (Sigma). Immunodetection was performed with antibodies directed against BRCA2 (Calbiochem, OP95, 1:1000), BRCA1 (Cell Signaling, 9010, 1:1000), RAD51 (GeneTex, gtx70230, 1:1000), PARP1 (Cell Signaling, 9532, 1:1000), RAD52 (Santa Cruz, sc-365341, 1:250), SLX4 (BTBD12; Novus Biologicals, NBP1-28680, 1:1000), MUS81 (Abcam, ab14387, 1:1000), ERCC1 (Cell Signaling, 3885, 1:1000), HSP90 (Santa Cruz, sc-1055, 1:1000), and beta-Actin (MP Biomedicals, 69100 1:10000). Horseradish peroxidase (HRP)-conjugated secondary antibodies (DAKO) were used for visualization using chemiluminescence (Lumi-Light, Roche Diagnostics) on a Bio-Rad bioluminescence device, equipped with Quantity One/ChemiDoc XRS software (Bio-Rad).

## Strand-seq library preparation and sequencing

Strand-seq libraries were prepared as previously described[32,33], with a few modifications. Prior to sorting single cells, KB2P3.4 and KB2P4.4R3 were treated with 1 μM olaparib and KBM-7 with 0.15 μM olaparib for 48 h. To incorporate BrdU during one cell cycle, BrdU (Invitrogen) was added to exponentially growing cell cultures at 40 μM final concentration. Timing of BrdU pulse was 16 h for KB2P3.4R3 and KBM-7 cells, and 20 h for KB2P3.4 cells. After BrdU pulse, cells were resuspended in nuclei isolation buffer (100 mM Tris-HCl pH 7.4, 150 mM NaCl, 1 mM CaCl2, 0.5 mM MgCl2, 0.1% NP-40, and 2% bovine serum albumin) supplemented with 10 μg/ml Hoechst 33258 (Life Technologies) and propidium iodide (Sigma Aldrich). Single nuclei were sorted into 5 μl Pro-Freeze-CDM NAO freeze medium (Lonza) supplemented with 7.5% dimethyl sulfoxide, in 96-well skirted PCR plates (4Titude), based on propidium iodide and Hoechst fluorescence intensities using a FACSJazz cell sorter (BD Biosciences). For each experiment, 96 libraries were pooled and 250–450 bp-sized fragments were isolated and purified. DNA quality and concentrations were assessed on the Qubit 2.0 Fluorometer (Life Technologies) and using the High Sensitivity dsDNA kit (Agilent) on the Agilent 2100 Bio-Analyzer. Single-end 50 bp sequencing reads from the Strand-seq libraries were generated using the HiSeq 2500 or the NextSeq 500 sequencing platform (Illumina).

## Detection and mapping of breakpoints

Indexed bam files were aligned to mouse (GRCm38) or human genomes (GRCh38) using Bowtie254. Different R-based packages were used for the detection and mapping of breakpoints: Aneufinder2 was used for libraries with arbitrary copy number profiles (KB2P3.4 and KB2P3.4R3), while HapSCElocatoR (https://github.com/daewoooo/HapSCElocatoR) was used for libraries derived from the haploid cell line KBM-7. Aneufinder2 was used to locate and classify any type of breakpoint, not only template strand switches, using standard settings[100]. In short, copy numbers for both the Watson (negative) and Crick (positive) strand were called and breakpoints were defined as changes in copy number state. These breakpoints are then refined with read-resolution to make full use of the sequencing data. As Aneufinder2 also detects stable chromosomal rearrangements, clonal aberrations were defined as events that occurred at the exact same locations in > 25% of the libraries from one cell line. HapSCElocatoR is implemented in the R package fastseg[101], and uses circular binary segmentation to localize SCEs in haploid Strand-seq libraries as a change in read directionality from Crick to Watson or vice versa. Only non-duplicate reads with a mapping quality greater than or equal to 10 were analyzed. We considered only strand state changes with at least three directional reads on both sides of the putative SCE site as an SCE event. Single directional reads embedded within an extended region with the opposite directionality were considered as errors and their directionality was flipped. Computationally localized SCE or somatic copy number alteration (SCNA) events were further manually verified by visual inspection of chromosome ideograms (obtained from Aneufinder2 or BAIT; see Figs. 1A and 5A respectively).

## Detection and analysis of SCE hotspots

HapSCElocatoR-generated '.bed'-files containing the locations of all mapped SCE events were uploaded to the UCSC Genome Browser and hotspots were identified as regions containing multiple overlapping SCEs. p-values were assigned to putative SCE hotspots using a custom R-script, based on capture–recapture statistics. Briefly, the genome was divided into bins of the same size as the putative hotspot and the chance of finding the observed number of SCEs in one bin was calculated based on the total number of SCEs detected in the cell line.

## Genomic analysis of SCE localization

A custom Perl script was used for the permutation model (https://github.com/Vityay/GenomePermute)[34]. For each of 1000 permutations, a random number $n$ was generated and all SCEs were shifted downstream by $n$ bases on the same chromosome. To prevent small-scale local shifts, n was confined to be a random number between 2 and 50 Mb. When the resulted coordinate exceeded chromosome size, the size of chromosome was subtracted, so that the SCE is mapped to beginning part of the chromosome, as if the chromosome was circular. All annotated assembly gaps were excluded before our analysis, to prevent permuted SCE mapping to one of the gap regions. The number of SCEs overlapping with a feature of interest in each permutation was then determined, as well as the original SCE regions. All values were normalized to the median permutated value, in order to determine relative SCE enrichments over expected, randomized distributions and to allow for comparison of the different cell lines. Significance was determined based on the amounts of permutations that showed the same or exceeding overlap (enrichment) or the same or receding (depletion) overlap with a given genomic feature compared to overlap between the original SCEs and the same feature. Any experimental overlap that lies outside of the 95% confidence interval found in the permutations has a p-value below 0.05 and was deemed significant. Experimental overlaps lying outside of the permuted range were given a p-value below 0.001, as there was a < 0.1% (1/1000) chance of such an overlap occurring by chance.

Enrichment analyses for G4 motifs were performed using a 10 Kb SCE region size cutoff. Putative G4 motifs were predicted using custom Perl script by matching genome sequence against following patterns: G3 + N $x$ G3 + N $x$ G3 + N $x$ G3+, where $x$ could be the ranges of 1–3, 1–7, or 1–12 bp. Enrichment analysis for coding genes, CFSs[49], centromeres, and telomeres were performed using a 100 Kb size cutoff. Genome and gene annotations were obtained from Ensembl release 88 (GRCh38 assembly, http://www.ensembl.org). Gene bodies were defined as regions between transcription start sites and transcription end sites.

## Flow cytometry

RPE-1 cells were treated with 20 μM EdU for 0, 8, 12 or 24 h, subsequently fixed in ice-cold ethanol (70%) for at least 16 h, and stained with primary antibody against phospho-histone-H3-Ser10 (Cell Signaling; 9701, 1:100) and Alexa-488-conjugated secondary antibodies (1:200). EdU Click-it reaction was performed with Alexa-647 azide according to the manufacturer's instructions (InvitrogenTM). DNA was stained using propidium iodide following RNase treatment. At least 10,000 events per sample were analyzed on an LSR-II flow cytometer (Becton Dickinson, Franklin Lakes, NJ, USA). Data were analyzed using flowjo software (Becton Dickinson).

## Xenopus laevis egg extracts and biotin-oligonucleotide pull-downs

Cytostatic factor (mitotic) and low speed supernatant (LSS) extracts were prepared according to Murray and Blow respectively[102,103]. Biotin-oligonucleotide pull-down MS was performed as previously described[104]. In short, a biotinylated-oligo (5′- A*CGCTGCCGAATTCT ACCAGTGCCTTGCTAGGACATCTTTGCCCACCTGCAGGTTCACCC-3′, *=biotin) was annealed to its reverse complement at a concentration of 10 μM in 50 mM Tris pH8.0 buffer. The oligo-duplexes were diluted to 100 nM, after which 10 μl oligo was coupled to 60 μl streptavidin-coupled magnetic beads (Dynabeads MyOne Streptavidin C1, Invitrogen) by incubation for 60 mins in wash buffer I (50 mM Tris pH 7.5, 150 mM NaCl, 1 mM EDTA pH8.0, 0.02% Tween-20). Excess oligo-duplexes were removed by three washes in IP buffer (ELB-sucrose buffer: 10 mM HEPES-KOH ph7.7, 50 mM KCl, 2.5 mM MgCl$_2$, 250 mM sucrose; 0.25 mg/mL BSA; 0.02% Tween-20), after which the oligo-beads mixture was suspended in 40 μl IP buffer. Mitotic and interphase extracts were thawed on ice from −80 °C and supplemented with 20x

energy mix (20 mM ATP, 150 mM Creatine Phosphate, 20 mM MgCl$_2$, 2.5 mM EGTA). For biotin-oligonucleotide pulldown 8 µl mitotic or interphase extract was incubated with 4 µl of oligo-beads mixture for 10 mins. Beads-extract mixture was washed two times with 400 µl of IP Buffer, two times with IP-buffer minus BSA, and lastly one time with ELB-sucrose buffer. After the final wash, beads were taken up in 50 µl denaturing buffer (8 M Urea, 100 mM Tris ph8.0) and snap frozen. Mass spectrometry of oligonucleotide-bound proteins was performed by on-bead digestion as previously described for plasmid pull-down MS[105]. Two biological replicate experiments were performed, each with three technical replicate measurements per sample.

## Mass spectrometry

Online chromatography of the extracted tryptic peptides was performed using an Ultimate 3000 HPLC system (Thermo Fisher Scientific) coupled online to a Q-Exactive-Plus mass spectrometer with a NanoFlex source (Thermo Fisher Scientific), equipped with a stainless-steel emitter. Tryptic digests were loaded onto a 5 mm × 300 µm internal diameter (i.d.) trapping micro column packed with PepMAP100, 5 µm particles (Dionex) in 0.1% formic acid at the flow rate of 20 µl/minute. After loading and washing for 3 min, trapped peptides were back-flush eluted onto a 50 cm × 75 µm i.d. nanocolumn, packed with Acclaim C18 PepMAP RSLC, 2 µm particles (Dionex). Eluents used were 100:0 H$_2$O/acetonitrile (volume/volume (V/V)) with 0.1% formic acid (Eluent A) and 0:100 H$_2$O/acetonitrile (v/v) with 0.1% formic acid (Eluent B). The following mobile phase gradient was delivered at the flow rate of 250 nl/min: 1–50% of solvent B in 90 min; 50–80% B in 1 min; 80% B during 9 min, and back to 1% B in 1 min and held at 1% A for 19 min which results in a total run time of 120 min. MS data were acquired using a data-dependent acquisition (DDA) top-10 method, dynamically choosing the most abundant not-yet-sequenced precursor ions from the survey scans (300–1650 Th) with a dynamic exclusion of 20 s. Survey scans were acquired at a resolution of 70,000 at mass-to-charge (m/z) 200 with a maximum inject time of 50 ms or AGC 3E6. DDA was performed via higher energy collisional dissociation fragmentation with a target value of 1x10E5 ions determined with predictive automatic gain control in centroid mode. Isolation of precursors was performed with a window of 1.6 m/z. Resolution for HCD spectra was set to 17,500 at m/z 200 with a maximum ion injection time of 50 ms. Normalized collision energy was set at 28. The S-lens RF level was set at 60 and the capillary temperature was set at 250 °C. Precursor ions with single, unassigned, or six and higher charge states were excluded from fragmentation selection. Statistics analysis was conducted in Perseus using two-sided unpaired Students' T tests with equal variance. A false discovery rate (FDR) of 0.01 was used to indicate significant hits.

## Live cell microscopy

RPE-1 *TP53*$^{-/-}$ shBRCA2 cells were seeded in 8-well cover glass chambers (Lab-Tek-II, Nunc) at 50% confluency. 48 hours prior to plating, cells were treated with doxycycline (0.1 µg/mL). 16 hours prior to imaging, olaparib (0.5 µM) was added where indicated. Novobiocin (POLQi) was added at the start of imaging at a final concentration of 50 µM. DIC images were obtained every 7 minutes over a period of 10 hours using a Nikon Eclipse Ti-E inverted microscope, equipped with a Hamamatsu C11440-22CU digital camera, and 12 V/100 W halogen lamp. In the Z-plane, 5 images were acquired at 1-micron interval. Image analysis was performed using NIS-Elements software.

## Electron microscopy analysis of DNA intermediates

Electron microscopy (EM) analysis was performed according to the standard protocol[24,106], with modifications. For DNA extraction, cells were lysed in lysis buffer and digested at 50 °C in the presence of Proteinase-K for 2 h. The DNA was purified using chloroform/isoamyl alcohol and precipitated in isopropanol and given 70% ethanol wash and resuspended in elution buffer. Isolated genomic DNA was digested

with PvuII HF restriction enzyme for 4 to 5 h. DNA was washed with TE buffer and concentrated using Amicon size-exclusion column. The benzyldimethylalkylammonium chloride (BAC) method was used to spread the DNA on the water surface and then loaded on carbon-coated nickel grids and finally DNA was coated with platinum using high-vacuum evaporator MED 010 (Bal Tec). Microscopy was performed with a transmission electron microscope FEI Talos, with 4 K by 4 K cmos camera. Images were processed and analyzed using the MAPS software (FEI) and ImageJ software.

## Analysis of whole-genome sequence data

Deletion calls were downloaded from the ICGC Data Portal (https://dcc.icgc.org/releases/release_28/Projects/BRCA-EU). HRD status, as previously assessed by the CHORD algorithm, was acquired from Nguyen et al[57]. *BRCA1/2* mutation status was obtained from Davies et al[16]. The coordinates of common fragile site were acquired from Georgakilas et al[49]. FRA1C, FRA5D and FRA13C were excluded as they were fully enveloped within another CFS. Furthermore, only autosomal deletions and CFS were considered. A deletion was only considered as being in a CFS if the entire deletion was positioned within a single CFS. Microhomology was determined by calculating the length of consecutive overlapping nucleotides from the 3' and 5' deletion breakpoints. Microhomology of larger or equal to 2 bp was used for analysis.

## Reporting summary

Further information on research design is available in the Nature Portfolio Reporting Summary linked to this article.

## Data availability

The mass spectrometry proteomics data generated in this study have been deposited in the ProteomeXchange Consortium via the PRIDE partner repository with identifier PXD028670. The Strand-seq data have been deposited in the European Nucleotide Archive (ENA) at EMBL-EBI under accession number PRJEB47697. Source data are provided with this paper. Uncropped Western Blots for Fig. 2a are not available. Source data are provided with this paper.

## Code availability

HapSCElocatoR can be found at https://github.com/daewoooo/HapSCElocatoR.

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

## Acknowledgements

This work was supported by grants from the Netherlands Organization for Scientific Research (NWO-VIDI 917.13334 to M.A.T.M.v.V. and Gravitation program 'CancerGenomiCs' to P.K.), the European Research Council (ERC-Consolidator grant #681572 'TENSION' to M.A.T.M.v.V.), ERIBA-UMCG funding to A.M.H., P.L. and M.A.T.M.v.V. We thank Jos Jonkers, Thijn Brummelkamp and Shunichi Takeda for sharing reagents. We thank members of the Medical Oncology Department and ERIBA for feedback on the manuscript.

## Author contributions

A.M.H., P.L., and M.A.T.M.v.V. conceived the project. A.M.H., C.S., Y.P.K., M.E., R.H.d.B., A.A., F.J.B., and E.W. conducted cell biological experiments. A.A., R.H.d.B., and P.K. coordinated and performed mass spec analyses using *Xenopus* egg extracts. M.T. provided reagents. D.C.J.S. coordinated sequencing analyses. D.P. and V.G. performed bioinformatics analysis. J.K.d.K., M.T., and R.v.B. analyzed WGS tumor data. E.M.M. and A.R.C. conducted and analyzed ssDNA and EM analyses. A.M.H., C.S., and M.A.T.M.v.V. wrote the manuscript. All authors provided feedback on the manuscript.

## Competing interests

M.A.T.M.v.V. has acted on the scientific advisory board of RepareTx, which is unrelated to this work. The other authors declare no competing interests.

## Additional information

[1]Department of Medical Oncology, University Medical Center Groningen, University of Groningen, the Netherlands, 9713GZ Groningen, the Netherlands. [2]European Institute for the Biology of Ageing, University Medical Center Groningen, University of Groningen, 9713GZ Groningen, the Netherlands. [3]Department of Genome Sciences, University of Washington School of Medicine, Seattle, WA 98195, USA. [4]Department of Molecular Genetics, Erasmus MC Cancer Institute, Erasmus University Medical Center, 3000 CA Rotterdam, the Netherlands. [5]Princess Máxima Center for Pediatric Oncology, 3584 CS Utrecht, the Netherlands. [6]Oncode Institute, 3521 AL Utrecht, the Netherlands. [7]Department of Human Genetics, Leiden University Medical Center, 2333 ZC Leiden, the Netherlands. [8]Hubrecht Institute-KNAW and University Medical Center Utrecht, Utrecht, the Netherlands. [9]Terry Fox Laboratory, BC Cancer Agency, Vancouver, BC V5Z 1L3, Canada. [10]Department of Medical Genetics, University of British Columbia, Vancouver, BC V6T 1Z4, Canada. [11]Present address: Department of Genetics, University Medical Center Groningen, University of Groningen, the Netherlands, 9713GZ Groningen, the Netherlands. [12]Present address: MRC Human Genetics Unit, Institute of Genetics and Cancer, The University of Edinburgh. Crewe Road South, Edinburgh, EH4 2XU Edinburgh, UK. [13]These authors contributed equally: Anne Margriet Heijink, Colin Stok. ✉e-mail: plansdor@bccrc.ca; m.vugt@umcg.nl

