## [Peer Review File · Nature Communications]

Sister chromatid exchanges induced by perturbed replication can form independently of BRCA1, BRCA2 and RAD51Editorial Note: This manuscript has been previously reviewed at another journal that is not operating a transparent peer review scheme. This document only contains reviewer comments and rebuttal letters for versions considered at *Nature Communications* .

REVIEWERS' COMMENTS

Reviewer #1 (Remarks to the Author):

This manuscript proposes that sister chromatid exchanges (SCEs) induced by perturbed replication are formed independently of homologous recombination (HR) factors.

General Comment

My overall impression for the original version was that the authors had performed a large body of high-quality work that was suitable for publication in Nature Communications. In the rebuttal, the authors have to a large degree addressed my concerns. The model shown in Fig 5F is quite interesting and has novel features and my recommendation is that the manuscript should be accepted for publication, once the authors describe a little better the results shown in Figs 3C and 3D (see specific comment below). I also note that the inclusion of the data shown in Fig 6 supports the relevance of the findings in human cancers, as microhomologies are frequently observed at sites of chromosomal rearrangements in human cancers.

Specific Comment

The authors report that there is an enrichment of SCEs in common fragile sites (CFSs) in cells treated with olaparib and that this enrichment is independent of BRCA2 status. However, Fig 3C seems to show that most of the SCEs are in olaparib-treated BRCA2-deficient cells. The statistical analysis (which is complex and hard to follow) in Fig 3D does not support the visual presentation of the data shown in Fig 3C. In their rebuttal, the authors did not address my concern that the BRCA2 status may affect the presence of SCEs at CFSs. It could very well be that SCEs at CFSs may behave differently than SCEs at the rest of the genome: thus, the SCEs at CFSs may be more frequent when BRCA2 is depleted, whereas the SCEs at the rest of the genome may be equally frequent. If so, SCEs at CFSs would behave similarly to ssDNA gaps at replication forks (Fig 2B, the ssDNA gaps are more frequent in olaparib-treated BRCA2-deficient cells than in olaparib-treated BRCA2-proficient cells). Can the authors address this point in the Discussion of the manuscript and explain better the statistical analysis shown in Fig. 3D?

Reviewer #2 (Remarks to the Author):

The authors have addressed all my concerns and followed many of my suggestions. I recommend to go forward with publication.

Reviewer #1 (Remarks to the Author):

This manuscript proposes that sister chromatid exchanges (SCEs) induced by perturbed replication are formed independently of homologous recombination (HR) factors.

General Comment: *‘My overall impression for the original version was that the authors had performed a large body of high-quality work that was suitable for publication in Nature Communications. In the rebuttal, the authors have to a large degree addressed my concerns. The model shown in Fig 5F is quite interesting and has novel features and my recommendation is that the manuscript should be accepted for publication, once the authors describe a little better the results shown in Figs 3C and 3D (see specific comment below). I also note that the inclusion of the data shown in Fig 6 supports the relevance of the findings in human cancers, as microhomologies are frequently observed at sites of chromosomal rearrangements in human cancers.’*

Reply: We thank the reviewer for the constructive feedback, and we appreciate that the reviewer acknowledges the relevance of the findings, based on the analysis of cancer genomes.

Specific Comment: *‘The authors report that there is an enrichment of SCEs in common fragile sites (CFSs) in cells treated with olaparib and that this enrichment is independent of BRCA2 status. However, Fig 3C seems to show that most of the SCEs are in olaparib-treated BRCA2-deficient cells. The statistical analysis (which is complex and hard to follow) in Fig 3D does not support the visual presentation of the data shown in Fig 3C. In their rebuttal, the authors did not address my concern that the BRCA2 status may affect the presence of SCEs at CFSs. It could very well be that SCEs at CFSs may behave differently than SCEs at the rest of the genome: thus, the SCEs at CFSs may be more frequent when BRCA2 is depleted, whereas the SCEs at the rest of the genome may be equally frequent. If so, SCEs at CFSs would behave similarly to ssDNA gaps at replication forks (Fig 2B, the ssDNA gaps are more frequent in olaparib-treated BRCA2-deficient cells than in olaparib-treated BRCA2-proficient cells). Can the authors address this point in the Discussion of the manuscript and explain better the statistical analysis shown in Fig. 3D?’*

Reply: We would like to thank the reviewer for their thorough analysis of our SCE enrichment analysis in Figure 3D. We have re-analyzed the overlap between SCEs and CFSs using a different (curated) set of CFS genomic locations (Georgakilas et al., 2014, PMID: 25238782). Using this new analysis, we were able to more carefully analyze overlap, and we obtained slightly stronger enrichment for olaparib-induced SCEs at CFSs in BRCA2-deficient cells compared to BRCA2-proficient cells. These results align well with the examples presented in figure 3C, and agree with the statement of the reviewer that BRCA2-status may affect the number of SCEs at CFSs. The details on our statistics methodology have been published previously (Van Wietmarschen *et al.*, 2018, PMID: 29348659), which we refer to in our methods section. We cannot rule out a link between ssDNA at replication forks and SCE

formation, we do believe this is an interesting point of discussion. To address this issue, we have included the following lines in the discussion: *'The enrichment of both olaparib-induced SCEs and ssDNA gaps in BRCA2-deficient cells, warrants further research into potential shared mechanisms between SCE and ssDNA gap formation at stalled forks. If SCE formation at CFSs is directly linked to stalled replication forks, this would suggest that SCEs at CFSs may arise through different mechanisms than SCEs elsewhere in the genome.'* (Lines 297-301).